# From the Lab Notebook: Observations on Tactile Sensing for Robotic Manipulation

Julia Di

Mechanical Engineering

Stanford University

Stanford, California 94305

Email: juliadi@stanford.edu

*Abstract*—**Manipulation is having a moment. Over the last decade, academic and commercial players alike have renewed interest in robotic manipulation due to a variety of factors: increasing availability of robotic arms, standardization of grippers, and decreasing cost of compute power. One of the key challenges that remain is making manipulators robust and safe for interaction in unstructured environments. Tactile sensors, by augmenting visual perception with shape, texture, compliance, temperature, vibration, and/or force feedback, are a vital step towards making robotic manipulators useful outside the lab sandbox. As a reflection on tactile sensing for robotic manipulation, this paper has two goals: 1) to provide practical advice on choosing a tactile sensor design, and 2) to discuss prevalent sensing archetypes and trends through a historical lens. We conclude this paper with a discussion of future research questions and community goals given trends in tactile sensor design.**

## I. INTRODUCTION

We take for granted how easy dexterous manipulation is. Even routine tasks, such as picking up a dirty glass to load the dishwasher, are fundamentally difficult manipulation problems. Our fingertips effortlessly convey information about the weight, temperature, and slipperiness of the glass, so that it does not slip from too little force, or break under too great force. Our skin easily senses incidental contacts, so that our hands can intelligently move in restricted workspaces like a dishwasher without knocking over other dishes. Take away the feedback from our mechanoreceptors, and even simple object manipulation is difficult [3, 21].

For a robotic manipulator to perform the same type of task with human-like performance, it must have similar sensory feedback—vision alone does not provide sufficient force or spatial resolution for the cluttered and constrained environments of daily life [4, 6, 36]. Many tactile sensors have been published over the decades since the field began; a quick Google search reveals countless commercial and academic sensory options for transducing basic physical properties. Although many review papers thoroughly compare and contrast tactile fabrication, transduction, and computation methods to great detail, there are few practical guides for tactile sensors in manipulation. In this work we aim to provide high-level direction, advice, and historical context for tactile sensors and sensing trends, and discuss emerging themes for the future.

## II. CRASH COURSE IN TACTILE SENSORS

In this section, we aim to give the reader a high-level understanding of practical considerations, and a black-box description of tactile sensors. This paper does not aim to be a comprehensive review of tactile sensing as a field; for that purpose, we ask readers to refer to review papers such as [8, 22, 41]. But having a general understanding of the inputs, transduction methods, and outputs of tactile sensors, as well as the engineering concerns with using them, are important to selecting a sensor for one's specific manipulation problem design criteria.

First, tactile sensing can be defined as the transduction of some information or property of an object through physical contact between the sensor and the object [41]. Many tactile sensors are based on arrays of sensing elements called taxels, where each taxel provides one signal.

Next, we discuss high-level design parameters for tactile sensors. Though rarely mentioned in research publications, engineering concerns should always be kept in mind during the design process. Some practical rules of thumb for tactile sensors for robotic manipulation are that they should be:

- **Realistic**. A tactile sensor for an end-effector should be realistically mountable and useable. This usually translates to being *low cost*, requiring *low energy*, and having an easy assembly and integration process, such as using a *low number of wires* (as integrating many connections will make robotic end-effectors bulky and affect their workspace). Similarly, the electrical connections must be robust given the type of movements required of the manipulation task, as well as the complexity of supporting circuitry and software. Robust electrical connections are especially important for soft sensors, which must inevitably connect back to a rigid circuit board.

- **Perceptive**. Sensors should be designed to detect the desired property over the desired area. Most sensors focus on point contacts (usually measuring contact force) at the fingertips, in which case the sensor design must be well integrated with the finger [13]. Some sensors are for larger areas of contact, such as the back of a robotic hand, in which case they should be flexible and/or stretchable for robotic hands with larger degrees of freedom.

TABLE I: Comparison of common transduction methods for sensors in robotic manipulation

| Transduction method | Pros | Cons | Examples |
|---|---|---|---|
| Capacitance | Sensitive, good spatial resolution, large dynamic range, some commercial options, can be flexible and stretchable | Complex circuit design (minimizing crosstalk and stray capacitance), susceptible to noise, temperature, nonlinearity, and hysteresis | RoboTouch Digitact [1], iCub capacitive sensors [33] |
| Piezoresistance | Simple, low cost, widely available commercially, can be flexible | Temperature and moisture dependent, susceptible to fatigue, high power consumption | Scalable Tactile Array Glove (STAG) [35] |
| Barometric | High sensitivity, temperature independent, can be flexible | Low spatial resolution, handling constraints | BioTac (commercial fluid barometer and thermistor) [14] [34] |
| Optical | High spatial resolution, repeatable, integrates well with modern data-driven techniques, can be flexible | Bulky, high power consumption, usually nonlinear, high computational costs | GelSight [43] |

- **Interpretable**. The sensor should be interpretable to be useful. Traditionally, tactile sensor designers used *linearity* as a metric of interpretability, but more recent high resolution non-linear sensors are made interpretable through data-driven techniques such as deep neural networks, which will be discussed in Section III-B. Sensor calibration is a related factor.
- **Suitable**. All sensors must be chosen with consideration for the performance metrics of the task: these usually include the measurement range, hysteresis, signal-to-noise ratio, and repeatability / reliability [11]. There is usually a trade-off between sensitivity, spatial resolution, frequency response, and dynamic range. The prioritization between these design parameters depends on the task at hand. For example, tasks that require fine texture recognition will prioritize high spatial resolution over sensitivity, frequency response, and dynamic range [44].

Next, we describe tactile sensors as systems, and begin with the inputs into tactile sensors. Generally, researchers desire to understand either the contact condition or the object of interest. Contact-related inputs include contact forces and torques, locations of contact forces, pressure distributions, joint angles (e.g. for proprioception) and vibrations (e.g. for slip detection). Object-related inputs include the object's shape, texture, compliance, and temperature.

Tactile sensors can be roughly classified by the way they transduce a given input. Common transduction methods are given briefly in Table I along with examples. Recent surveys [22, 41] provide in-depth reviews of a broader set of transduction methods.

Finally, we discuss the outputs of tactile sensors by connecting them to their applications. In robotic manipulation, tactile sensors are generally used to characterize and identify object properties, provide reactive context for events, and/or close a control loop [36]. Traditional applications for tactile sensing related to object properties include: texture recognition, contact shape recognition, temperature recognition, and object classification. Traditional applications for tactile sensing related to reactive context and control include: grasp force control, grasp stability assessment, slip recognition and detection, collision detection, and tactile servoing.

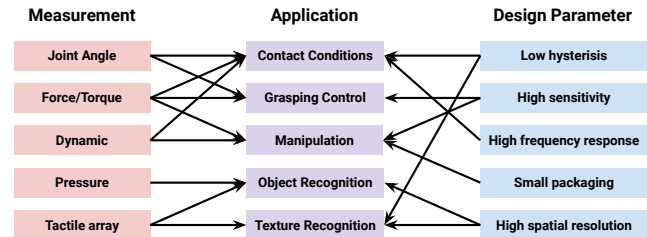

Fig. 1: Mapping tactile measurements and design parameters to broad applications in robotic manipulation.

Figure 1 provides a broad summary of tactile sensors as sensing systems, from input measurements to application areas, and notes the relevant general design parameters.

## III. TRENDS IN TACTILE SENSOR ARCHETYPES

Viewing tactile sensors as a basic black box system (input, transduction method, and output) allows a better understanding of how tactile sensors are used to further research trends in manipulation. In this section, we provide a historical perspective on tactile sensor designs to connect prevalent robotic manipulation problems and techniques to directions in sensor archetypes. Our aim is to provide an opinion of how sensor designs have changed over time, and as well as a first-order guide to which types of tactile sensing systems work in which applications of interest today. We conclude with Table II, which compares some common commercially-available tactile sensors and their usages.

### A. Origins: Single-Point Fingertip Sensors

Most robotic manipulators have tactile sensors located at the fingertips. While this paradigm has anthropomorphic origins (as the fingertips are among the most sensitive in the human hand [21]), it also stems from the fact that historically manipulation problems have been framed in terms of a gripper grasping an object with contact at its fingertips.

Grasping is one of the foundational skills in manipulation; how can one manipulate without grasping first? In the past, tactile sensors in grasping were concerned with measuring geometric parameters, such as the location of contact between the robot finger and the net force at the contact, in order to use

TABLE II: Comparison of Common commercially available tactile sensors for manipulation

| Commercial Sensor | Data types and usage | Pros | Cons |
|---|---|---|---|
| ATI Nano | Force/torque sensor for slip detection, force control | Typically considered benchmark standard for single-point force/torque sensing, limited computation needed | Expensive, fragile |
| Pressure Profile Systems (PPS) DigiTacts [1] | Capacitive pressure arrays for object recognition and classification [30] | Bluetooth (wireless), embedded electronics | Low spatial resolution compared to optical systems |
| BioTac [34] | Pressure and temperature sensing for slip detection, force control, object compliance measurements | Biomimetic multimodal sensing, integrated into fingertip | Expensive, only for fingertips |
| TakkTile sensors | MEMS barometric pressure arrays for slip detection, and force control | Can be flexible [20], durable, sensitive, adapted into Robotiq grippers | Low spatial resolution |
| Gelsight [43] | Tactile images for slip detection, object compliance measurements, texture recognition and classification | High spatial resolution, captures micro-patterns in image, slimmer variants available [12] | High computation costs, elastomer wear and tear |

the equations of motion to determine grasping behavior [19]. Common inputs measured to achieve those goals included joint angle sensors, together with a kinematic model to determine contact location, and force-torque information at the fingertip.

Nowadays, instead of determining how to grasp, most research in grasping is focused on grasp quality, most commonly grasp stability. In an industrial environment such as a factory, grasp quality is a simpler problem because generally object parameters are known. Any application in an unstructured environment, however, has strong incentive for tactile sensors to ensure a desirable, stable grasp, because object parameters are uncertain [18].

There are a few techniques for assessing grasp stability. A detailed technical review of them is beyond the scope of this paper and can be found in [5, 15]. However, at a high level, the most common techniques for grasping are:

1) estimating the friction cones at the fingertips, and using them as signals for slip [17, 27],
2) measuring the mechanical vibrations at the fingertips as signals for slip [37], and
3) measuring the current contact area of the fingertips through tactile arrays [42].

Historically, the development of the theory for friction models and friction cones [27] was the first major advance in grasp stability. The advantage of grasping with a friction model is that this is relatively fast and easy to compute with simple tactile sensors, and can predict (i.e. estimate) slip in advance; however, it does require a model of the friction between the fingertips and the object. The most common input of interest for estimating friction cones is force/torque input, to measure normal and tangential forces, or pressure input. Tactile sensor transduction methods varied from optical waveguides [26] to resistance-based pressure sensors [39].

Measuring vibrations directly only detects, and not predicts, the moment when slip occurs. Therefore, vibration-based approaches needed fast frequency response and thus only took off with the onset of high-speed data communication and processing. The advantage with vibration-based slip detection

is that one no longer requires a surface friction model, but the signal processing is more complex and introduces the need for protection from interference and noise. Vibrations are typically measured through dynamic tactile arrays using resistive, capacitive, and pressure sensor arrays [39, 14], and the high frequency and sampling rates are important engineering design parameters.

Once tactile sensor research scaled from single-point sensors to arrays of taxels, tactile sensors could be used to measure contact area by analyzing changes in features in tactile "images". The advantage of tactile images is the independence from surface friction and ability to detect incipient slip, however this comes at higher computation and processing costs. Contact area traditionally was measured through arrays of piezoresistive or capacitive sensors (with one taxel generally corresponding to one signal) [32, 38], or more recently through optical means such as a camera [42]. In the latter case, high spatial resolution is the important engineering design parameter.

### B. The Data Revolution: Emergence of High Spatial Resolution Sensors

The advent of machine learning has revolutionized many industries, robotics included. Naturally, to feed the data hunger, a new wave of tactile sensor designs now focus on high spatial resolution [25]. But implicit in this synergy between machine learning and high spatial resolution tactile sensors is a new perspective: because machine learning is suited to processing non-linear and "raw" data quickly and with little pre-processing, machine learning methods help relax some of the scalability assumptions for tactile sensors. Whereas previously sensors achieved resolution through *arrays* of taxels, limited by a one taxel per signal assumption so that humans could manually decipher meaning, machine learning now allows for fundamentally nonlinear and signal-rich optical sensors by replacing traditional signal processing.

These high-resolution nonlinear sensors, made interpretable with computational advances and aided by the miniaturization of electronics, are analogous to imaging devices. Leveraging

the existing research in feature extraction from visual images, feature extraction from tactile images have unlocked progress in areas such as object recognition, contact pattern recognition, and state estimation [25]. Perhaps one of the most popular commercially available vision-based sensors for fingertips is GelSight [43] with many other high-resolution optical fingertip systems following suit within the last two years [2, 12, 29, 31, 40], and most recently [24]. Such tactile sensors, with machine learning techniques, have been shown in tasks like slip detection [42], multimodal grasping[7], fabric texture recognition [44], and pose estimation and tracking [2].

In some ways, machine learning techniques are uniquely suited for fusing tactile data with visual data, as a way to map multidimensional and "raw" data without explicit models. Yet there are still numerous disadvantages. Data is both a blessing and a curse — large open source tactile datasets still do not exist, so many research datasets are small in example size and self-created [2]. It takes time and resources to curate these datasets, and they are not sensor agnostic, as there is no universal tactile image. Furthermore, tactile images are not the same as visual images, yet most researchers use model architecture pre-trained on visual image databases like ImageNet [7, 44], which may lead to errors in models. For a more comprehensive review on tactile perception, see [25].

It is important to note that tactile perception from high resolution sensors is an active research area that has come about from mutual advances in computer science (machine learning) and mechanical design (rapid fabrication and dense electronic packaging). This synergy also highlights the need for mechanical designers to be fluent in computation methods (to design nonlinear sensors that are machine interpretable) and for computer scientists to be fluent in mechanical design (to better understand contact conditions for building learning models).

### C. Towards Human Performance: Stretchability and Electronic Skins

As electronics become more and more dense and scalable, a natural question arises: when will we move tactile sensors beyond just the fingertips, and outfit entire areas of the robotic hand? For a tactile sensor to cover areas like the palm or back of the hand, they must be stretchable and/or flexible, and must tolerate the same impacts that one would expect the back of a hand or a palm to feel. This requirement of stretchability leads to a whole host of important sensor performance considerations such as wear rate, maximum stretching and saturation, and reliability. These issues, in combination with the wiring and data acquisition concerns inherent to large-area tactile sensors, are problems that current technology is only just beginning to address [22].

In literature, an artificial skin is often considered to have the following characteristics: 1) is flexible and stretchable, and 2) have multiple modes of tactile sensing. A limited number of tactile sensing skins have been shown in robotic manipulators [28] in the last decade, where most are based on arrays of taxels. A comprehensive review on tactile skins can be found in [10, 23].

Just as with high-resolution tactile sensors, machine learning has also introduced new possibilities in stretchable and flexible sensor design by freeing arrays from the requirement of *linearity* as signal interpretability. Most recently, deep convolutional neural networks (CNNs) were shown to interpret human grasp "signatures" from an array of 548 piezoresistive sensors that were assembled on a knitted glove [35], a staggeringly dense array that is difficult to interpolate through traditional linear processing means.

Given the exciting advances in tactile sensing, which has historically been in the realm of hardware engineering, it may be easy to feel that computation can "solve" manipulation. On the other hand, it is also easy to feel that machine learning is a distraction from first principles, which does not serve to advance our understanding of manipulation at a fundamental level. Regardless, the questions of tactile sensing design and tactile perception are quickly becoming intertwined, and future work requires cross-disciplinary researchers.

### IV. CONCLUSION

Tactile sensors augment perception beyond what vision can provide alone; perhaps the best example of this is how crucial manipulation information is for humans [21]. With the renewed interest in robotic manipulation, research in tactile sensor design and tactile perception has also been brought to the fore. Yet, most publications and presentations focus exclusively on the final product. Section II provides an instructive view on the factors of tactile sensor design and selection, and Section III discusses trends in tactile sensing from a historical perspective with emphasis on current directions. In the following section, we pose interesting future possibilities for tactile sensing given today's trends.

### V. DISCUSSION AND FUTURE DIRECTIONS

#### A. Tactile Exploration

In animals and humans, tactile sensing is used for manipulation, for exploration, and for reaction to external agents [9]. Yet in robotics, tactile sensing is used primarily for control in grasping and manipulation. As robots move from industrial manufacturing to tasks in unstructured environments, manipulators also take on a new role as *exploratory* instruments.

Let us revisit the kitchen example of Section I. When we reach into a sink, we need to obtain information about the state of objects, such as their locations and conditions (e.g. wet, slippery, dirty). Cameras are not able to provide that information in poor visual conditions, such as occlusion from clutter and poor lighting, but the hand still can.

Thus, tactile sensing is not just a means to provide closed loop control, but is also an exploratory methodology that improves the robot's ability to accomplish tasks. Interestingly, the trends in tactile sensing towards compliant (stretchable and flexible), light (low effective inertia at contact), high resolution, and robust sensors—while originally emphasized for the traditional control and classification goals of manipulation, are

also well-suited towards exploration and information acquisition beyond manipulation, and this would be an interesting research direction to pursue.

### B. Tactile Communication

Just as we take for granted the ease of dexterous manipulation, so too do we take for granted the cultural complexity embedded within the sense of touch. We can poke, prod, tickle, hug, and convey a myriad of emotions and social cues through a single type of stimuli. Located at the ends of arms, hands are not just tools for interaction but also for expression: a symbolic extension of the self in addition to a physical one.

While the human sense of touch is studied in the behavioral sciences [16], there has been relatively little attention on the importance of touch in human-robot systems. If robots are to be companions at home and coworkers in the office, able to help people do their dishes and reach the top shelf, they must utilize their hands in a way that is understandable to humans and conveys the desired meaning. Not coincidentally, the trend for large-area tactile skins is motivated by this goal.

## ACKNOWLEDGMENTS

The author is supported by a NASA Space Technology Research Fellowship.

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
