# OpenReview forum: "From the Lab Notebook: Observations on Tactile Sensing for Robotic Manipulation"
_roboticsfoundation.org/RSS/2020/Workshop/RobRetro — RobRetro 2020_

### Official Review · AnonReviewer1 · 2020-06-24
**A review on tactile sensor design**

**Confidence:** 4
**Rating:** 8

**Review:**

This work provides a 1) overview on design choices for tactile sensing and how they related to manipulation applications 2) a historic overview of existing tactile sensing designs (and their connection to advances in machine learning) 3) a discussion on future research directions to consider. Overall this makes this work a nice overview/retrospective on existing tactile sensing, with an interesting component of how advances in machine learning are connected to advances to touch sensing.

some recent work the authors might want to include: https://ieeexplore.ieee.org/abstract/document/9018215

---

### Decision · Program_Chairs · 2020-06-25

Accept